# Efficacy and Usability of eHealth Technologies in Stroke Survivors for Prevention of a New Stroke and Improvement of Self-Management: Phase III Randomized Control Trial

**DOI:** 10.3390/mps2020050

**Published:** 2019-06-13

**Authors:** Leire Ortiz-Fernández, Joana Sagastagoya Zabala, Agustín Gutiérrez-Ruiz, Natale Imaz-Ayo, Ander Alava-Menica, Eunate Arana-Arri

**Affiliations:** 1Osakidetza-Basque Health Service, Biocruces Bizkaia Health Research Institute, Cruces University Hospital, Plaza Cruces 12, 48903 Barakaldo, Spain; joana.sagastagoyazabala@osakidetza.eus (J.S.Z.); agusaguilar84@gmail.com (A.G.-R.); 2Biocruces Bizkaia Health Research Institute Plaza Cruces 12, 48903, Barakaldo, Spain; natale.imazayo@osakidetza.eus (N.I.-A.); juanandres.alavamenica@osakidetza.eus (A.A.-M.); eunatea@outlook.es (E.A.-A.)

**Keywords:** stroke, rehabilitation, self-management, quality of life, eHealth, decision support system, prevention

## Abstract

Background: Stroke is a leading cause of severe and long-term disability in developed countries. Around 15 million people suffer a stroke each year, being most of them ischemic due to modifiable risk factors. Adequate self-management abilities may help to manage the consequences of stroke, but it is unknown which specific intervention could be effective to booster these self-management abilities. Objective: To evaluate the improvement of self-management in chronic stroke survivors using decision support and self-management system (STARR). Methods: A randomized, prospective, parallel group, open, and the unicentric pilot trial will be performed. Stroke survivors and their caregivers will be randomly allocated to STARR management or standard of care. Main inclusion criteria are mild to moderate disabled first stroke adult survivor, living at home, able to cope and follow the guidelines and devices, without socio-familial exclusion. All will get a conventional treatment in the acute and subacute phase; however, in the chronic period, cases will use the developed STARR App and Decision Support System. Measurements will be performed at baseline, at 3 months, and at 6 months. Outcome measures are patient-report outcome measure of self-management competency, physical function, risk factor reduction, healthcare resource utilization, knowledge of the condition, mood, and social isolation. Discussion: If effective, the results of this study will enable stroke patients and their caregivers to deal better with the everyday life obstacles of stroke, improve the adherence of the treatment, improve the control of cardiovascular risk, and, in consequence, reduce the recurrence of secondary strokes, the number of complications, the number of consultations, and readmissions; to ultimately reduce the health systems costs. Taking into consideration that the number of stroke survivors is increasing around the world, a large number of individuals could profit from this intervention.

## 1. Background

Stroke is a leading cause of severe and long-term disability in developed countries, having an estimated total cost of approximately €64 billion per year in Europe for 2010 [1]. Around 15 million people suffer a stroke each year; [2] most of them being ischemic due to modifiable risk factors, such as hypertension, diabetes, obesity, dyslipidemia, smoking, alcohol consumption, sedentary life, and unhealthy diet [3]. Moreover, the number of persons having a stroke in Europe is likely to increase from 1.1 million per year in 2000 to more than 1.5 million per year in 2025, solely because of the aging population [4]. One-third of the people going through a stroke die. Thus, in Europe, there are around 650,000 stroke deaths each year [5]. One-third of the people who have suffered a stroke are left permanently disabled, with complications, including motor, cognitive, and language impairments, as well as psychological problems [6]. Estimates indicate that more than one-third of the patients still require assistance for daily living activities 6-years post-stroke [7]. It is well known that recurrent or secondary stroke carries with it a greater risk than the first-ever stroke for death and disability [8,9,10]. Fortunately, thanks to the new medical and technological advances in the treatment of acute stroke (e.g., mechanical thrombectomy and intravenous fibrinolysis), the mortality from the first stroke has decreased over the past years. However, consequently, the number of people at risk for a secondary stroke has increased, with an associated increase in healthcare costs [11]. In the US, from 1995–2005, the stroke death rate fell approximately 30%, and the actual number of stroke deaths declined approximately 14% according to U.S. Centers for Disease Control and Prevention [12]. It has been generally acknowledged for years that nonadherence rates for chronic illness regimens and for lifestyle changes are around 50%, even though the success of the medical treatment is largely determined by adherence. Adherence in long-term conditions can be improved by self-management interventions [13,14]. To ease and improve self-management conditions, there has been a boost for new technology-based intervention, which has mostly been studied in diabetes, asthma, and hypertension [15,16,17,18]. The use of mobile phone messaging, such as short message service and multimedia message service, has been used, as well as smartphones and tablets in telemedicine [15]. The prevention of stroke has been studied in some papers that propose different strategies for reducing cerebro-cardiovascular risk factors. Among these strategies, there is evidence for a phone-based computer aided prevention system [19] and an individualized coaching program executed by well-trained stroke nurses [20]. Several reviews have shown that interventions mediating the new eHealth technologies can reduce the risk of suffering a stroke episode, improving the control of risk factors; nevertheless, all of them conclude that new and well-designed studies are needed [15,21]. Therefore, in this study, we propose the development of a mobile phone/tablet application to be used at the patient’s home, which acts as a coach for the patient and the caregiver providing recommendations for lifestyles and medical treatments based on evidence, which are related to the risk of a new stroke episode.

## 2. Methods/Design

### 2.1. Primary Objective

The aim of the study is to evaluate the improvement of self-management at home in chronic stroke survivors using the decision support and self-management system for stroke survivors (STARR) (compared with standard of care). Improving the self-management means increasing physical function, controlling risk factors for a new cardiovascular event, improving quality of life, as well as other items, such as self-management behaviors, adherence to treatments indicated by the doctor, improving health care resource utilization, knowledge of the condition, mood, and social isolation. The alternative hypothesis would be that STARR management improves self-management, as well as the quality of life of stroke survivors.

### 2.2. Secondary Objectives

Improvement of the study participants’:Self-management.Level of independence in the activities of daily living (ADL).Quality of life.Adherence to home-based rehabilitation and pharmacological and non-pharmacological treatment.Need for caregivers (family, care providers…).Recurrent stroke and the complications related to stroke.A number of hospital readmissions, emergency care, and outpatient visits to the hospital and primary care centers.

Also, the study will evaluate:Cost-effectiveness of the developed system.Accessibility of the developed system.Sustainability of the developed system.User’s satisfaction (by users, we mean stroke survivors, caregivers, and professionals using the developed system).Possible adverse events that the system can cause on the study participants.

In addition, the efforts will be made to identify new variables related to the prognosis of stroke.

### 2.3. Participants

Inclusion criteria age 18 years to 80 years;having a diagnosis of first ischemic stroke within the past 6 months;hemiparesis with mild (91-99) or moderate (61-90) disability (Barthel Index, BI);with or without speech pathology but able to understand simple orders (Mississippi Aphasia Screening Test, >45);able to cope and to understand the guidelines to use the devices;the life expectancy of at least 12 months;no severe cognitive impairments (Montreal Cognitive Assessment, MoCA, >26);without medical comorbidities that could interfere with the home-rehabilitation program (for example, severe aortic stenosis, respiratory failure, severe osteoarthrosis);without socio-familiar dystocia (Gijon’s socio-familial evaluation scale (SFES) <14)without a basal functional situation >1 by a Modified Rankin Scale (MRS).

Exclusion criteriamedical comorbidities that could interfere with the home-rehabilitation program.refusal to sign the informed consent and participate in the study.

### 2.4. Study Design

We will perform a prospective, randomized, parallel group, open, and unicentric pilot trial. We will analyze the data from 40 patients between 18 and 80 years old who have had an ischemic stroke, and whose family or friend caregivers are followed at Rehabilitation Unit at Cruces University Hospital. Two groups will be formed. The control group will get conventional treatment from the acute to the chronic phase of stroke, and the intervention group will get a conventional treatment to the end of the subacute phase. However, in the chronic period, they will use the developed STARR System, as well as commercial wearables.

The main stages of stroke recovery are defined by temporal terms (acute, subacute, and chronic). The acute stage refers to the first 7-days-period of the stroke event. The subacute stage represents the patients from the first week till 6 months, dividing this period in two: early subacute (up to 3 months) and late subacute (from 3 to 6 months). The chronic stage is beyond 6 months when brain repair processes suggest that the majority of the recovery has occurred [22].

The STARR system is based on the following components:**Wearables and connected objects**: provide regular information about the evolution of certain risk factors (e.g., physical activity, blood pressure) without taking over users’ attention. The devices of the STARR system are composed of a tensiometer, a glucometer, a heart rate band, a balance, and a thermometer. All of them are commercial devices tested and with CE marking.**A decision support system (DSS)**: implements personalized advice, guidance, and follow-up for daily life activities of the stroke survivors by analyzing the information coming from the wearables and the connected objects, a number of predictive models, and user profiles. The DDS has a system of alarms that will guide patients in making decisions, with recommendations, such as modifying life habits, consulting with their general practitioner, or going to the emergency department. These alarms are based on clinical practice guidelines with proven evidence in the management of patients with stroke. In addition, the responsible doctor of the study participant will have access to a control panel in the health system and a mobile application, where the alarms of the patients can be managed.**Predictive models**: will be populated by risk assessment information provided by validated predictive models calculating stroke risk. The risk estimation done by these models will be complemented by information from a model of human motion analysis and guidance developed during the project using Kinect’s cameras and a created program algorithm, which has been found to be very useful for assuring continued engagement in physical activities in clinical and home settings. It will also be supported by the implementation of models of behavior change to capture individual variations and attitude changes over time. The key requirement behind the implementation of these models is to motivate self-management by encouraging self-awareness and trend-awareness in lifestyle in the sub-acute and chronic phases of stroke.**Self-management apps (DSS user interface)**: tools to inform and encourage stroke survivors to self-manage their condition. The STARR system will try to determine the user’s reason for non-adherence using a mobile phone app and an online lifestyle diary. The user will then automatically receive generated messages with persuasive, tailored content. The content will be different at different stages of the initiation and maintenance of health behavior.**Serious games**: promote physical activity and rehabilitation at home with suggested activities in serious games with a screen and a mini-bike.

The different components are currently being developed and will be adapted to fit specified scenarios and services.

### 2.5. Reason for Withdrawal From the Trial

Patients can withdraw from the study at any time and revoke informed consent, both stroke survivors and caregivers. In addition, they may be withdrawn if one of the following events occurs:DeathLoss of follow-upSevere disease of the principal caregiverAny other problems that, in the opinion of the research team, justify treatment withdrawal.

### 2.6. Recruitment

All the patients who have suffered a first ischemic stroke with a neurologic deficit and have been admitted in the Acute Stroke Unit of the Neurology Department of our Cruces University Hospital will be assessed by a Rehabilitation Medical Doctor (RMD) with expertise in neuro-rehabilitation. The assessment will be done in the first 48–72 h from the stroke. The survivors who need conventional rehabilitation treatment will start it within the first 72 h from the cerebrovascular attack unless they present a medical contraindication, such as hemodynamic instability or others.

The rehabilitation treatment will be at a maximum tolerated intensity, daily from Monday to Friday till the end of the hospitalization. Before the discharge of the patient, an RMD will reassess the discharge destination to home, rehabilitation hospital as inpatient, or nurse home care.

At 6 months after the stroke, all the survivors who fulfill the inclusion criteria and who are willing to take part in the research will be recruited, including their caregivers. Those who will not be included are patients without familial support, or a caregiver and the formal caregivers of stroke patients will not be included if the patient does not want to take part in the study. The sample will be divided into intervention and control groups in consecutive order (Figure 1).

Participants will be enrolled in the external consultation at neuro-rehabilitation at Cruces University Hospital in Spain. We will include eligible individuals admitted in consecutive order, to avoid bias, taking into account the inclusion and exclusion criteria. All the participants will be volunteers, having signed the informed consent. The researchers responsible for the recruitment of patients will assure that the informed consent is collected with all the rigor it requires.

The assignment to the control/intervention group will be made through a probabilistic computer algorithm with a ratio of 1: 1 between groups. The Bioinformatics and Statistics Platform of Biocruces Bizkaia Institute will be responsible for the creation of the algorithm using a simple allocation mechanism in permuted blocks of variable size. Opaque envelopes will be used to guarantee the concealment of the generated sequence. The randomization list will remain on the platform at all times, guaranteeing the concealment of the sequence of randomization.

Afterward, the devices will be installed at participants’ homes.

### 2.7. Protocol

The treatment assessment schedule is displayed in Table 1 and Table 2. All patients will be treated through regular practice, and the patients of the intervention group will be supported by the STARR platform.

### 2.8. Control Group

The controls will continue following the conventional home recommendations and the standard of care. They will be scheduled to carry out health care recommendations and medical and rehabilitation guidelines at home. The exercises will also be scheduled: aerobic exercises, resistance-potentiation, and stretching. Visits will be made every six months during the first year of follow-up and then annually.

### 2.9. Intervention Group

Cases will integrate into their daily routines the designed system (STARR App, DSS, and wearables), measuring at least once a day the vital signs and motor activity during outdoor activities and cycloergometer/pedal exercising. As medical doctors, we will promote a healthy diet, no alcohol or smoking consumption, weight control, blood pressure control, glycemic control, and complications control (e.g., pain, spasticity, falls, infections, etc.).

The designed application will have alarm systems, both for the patient and for the caregiver. The alarms can guide the patient through actions that can recommend to call or go to your primary care physician or go to the emergency services.

The commercial wearable devices will be chosen according to different characteristics: Price, Bluetooth connection, valid for iOS/Android, battery duration, and user’s opinions.

The commercial wearables should be easy to use, comfortable, cheap, and accurate in order to be accessible for the users.

The survivors and the caregivers will be trained about the use of the devices and also about the measures they have to take (e.g., weight, blood pressure, waist circumference) during a period of one week to 10 days.

In case of any technical problem, the user will have a telephone number to put in contact with technical support. If it’s not possible to solve the problem remotely, a technician will go to the participant’s home.

### 2.10. Outcome Measures

The primary clinical focus in this pilot trial is the evaluation of self-management of the stroke survivors. The proposed primary outcome will be measured by the validated Southampton Stroke Self-Management Questionnaire (SSSMQ) and Family needs of stroke patient questionnaire.

A series of secondary outcomes will be evaluated at different time points, such as the effectiveness of stroke self-management interventions. They rely upon the following items:-Physical function (PF) assessed by the modified BI and Lawton index.-Risk factor reduction (blood pressure, analytical profile -glycemia, HbA1c, lipids, weight, heart rate control, medication compliance).-Self-management behaviors by self-reported information on lifestyles: diet assessed by the Mediterranean diet assessment tool, exercise assessed by tracking with the wearables, smoking, and alcohol consumption.-Healthcare resource utilization by the information available in the public health system in Osakidetza, Cruces university hospital.-Knowledge of condition to assess whether there is a relationship between health literacy and control of cardiovascular risk factors, number of complications, number of recurrences, low adherence to pharmacological and non-pharmacological treatment, and self-management behaviors.-Mood and social isolation by Goldberg scale that measures anxiety and depression.-Stroke Self-management questionnaire (SSMQ).

Other secondary outcomes will be focused on the International Consortium for Health Outcomes Measurement (ICHOM), such as survival, disease control, and acute complications.

All the questionnaires and measurements for the achievement of the outcomes will be collected through face-to-face medical visits in the neuro-rehabilitation department.

### 2.11. Safety

Collection of Adverse Events (AE): only reported AE, either spontaneously by the patient or during interviews with the patient and/or the caregivers during the monitoring of the trial. AE will be collected as soon as the patient signs the Informed Consent up to 30 days after the last visit.

All AE should be documented in the patient’s medical history and in the clinical trial electronic data record.

AE is any adverse health effect in a patient or subject of a clinical trial, although it does not necessarily have a causal relationship to the treatment. An AE may, therefore, be any unfavorable and unintended sign (including an abnormal laboratory finding), symptom, or illness temporarily associated with the use of a research drug, whether or not it is related to the investigational medicinal product.

Adverse Reaction (AR) is any harmful and unintended reaction to a research drug/dispositive, regardless of the dose administered. In the case of an adverse reaction, there is a suspicion of a causal relationship between the investigational drug/dispositive and the adverse event.

Severe Adverse Event (SAE) and Severe Adverse Reaction (SAR) are an adverse event or adverse reaction, which, at any dose:causes the death of the patientthreatens the life of the patientrequires hospitalization or prolongation of patient hospitalizationcauses disability or permanent or major disabilityresults in a congenital anomaly or malformation

All AR will be monitored until their resolution, or at least 15 days after discontinuation of study (whichever occurs first) until the toxicity returns to a grade ≤1, or until the toxicity is considered irreversible.

### 2.12. Follow-up Period

All patients recruited for the research clinical trial (RCT) will be reassessed during external consultations at month 3 ± 1 week (1^st^ visit) and at month 6 ± 1 week (final visit).

All the participants will get the same reassessment at the event day (evaluation a), recruitment visit (evaluation b), 3 months later (evaluation c), and 6 months of follow-up (evaluation d) taking into consideration that not all the tools are administered at each time.

### 2.13. Sample Size

To obtain a power of 80% to detect statistically significant differences when comparing the null hypothesis—a bilateral chi-squared test for two independent samples—taking into account that the level of significance is 5% and assuming that the proportion of dependence in the control group is 41.5% [23] and the proportion in the intervention group is 15%, a total of 72 stroke survivors will be needed. Taking into account the expected percentage of dropouts of 10%, it will be necessary to recruit 80 patients. The dependence will be measured by SSSMQ and Family needs of stroke patient questionnaire.

### 2.14. Statistical Analysis

Each variable will be characterized using frequency distributions central tendency, such as the mean and median, and to determine variability standard deviation or the interquartile range according to its distributional characteristics.

Comparisons between groups and intra-groups will be performed using parametric tests provided that the distributional characteristics of the data meet the requirements; otherwise, non-parametric tests will be used.

Comparisons between two continuous variables will be performed using the Pearson or Spearman correlation, depending on the distributional characteristics.

The comparison of proportions between categorical variables will be carried out by means of a chi-squared test or its corresponding Fisher correction. The continuous variables will be compared using the comparison of the means (t-student test or ANOVA) if the sample distribution is normal, and if the distribution is not normal, then the non-parametric Mann-Whitney U test will be utilized.

The populations will be analyzed by intention to treat analysis and as stated in the protocol.

A micro-data collection form will be developed. The analysis of costs will be done from the perspective of the health system and direct costs. The data collected will serve to estimate the actual cost of procedure data under usual practice conditions and costs derived from the implementation of the new intervention in the different contexts under study.

The collection of these data, together with data on effectiveness and utility, will enable subsequent studies on economic analysis and cost-effectiveness/utility to help decision-making on which alternatives are most effective and efficient from the point of view of the health system.

The SPSS 23.0 program for Windows (SPSS Inc, Chicago, IL, USA) will be used. A p-value of 5% or lower is considered to be statistically significant. The statistical analysis will be blind to the intervention group.

### 2.15. Quality Control and Assurance

For ensuring the quality of the study, data will be collected on the adherence to the intervention, the inclusion, and follow-up of all patients, as well as monitoring and quality of data entry.

If there were to be suspicion of a serious violation of the trial protocol, it would imply a significant effect on:(a)the physical or psychological integrity or safety of patients during the trial, or(b)the values of the study. The study sponsor will be contacted as soon as possible. In any case, all violations will be notified to the relevant authorities in accordance with current legislation.

The protocol will not be modified without the consent of the principal investigator and the sponsor. Any modification of the protocol requires Ethics Committee approval prior to its implementation except to avoid immediate risks in patients.

The study documents will be reviewed, comparing them to the originals. The researcher will be informed of the progress of the study, and the suitability of the facilities will be evaluated on an ongoing basis.

In addition, the study may be evaluated by internal auditors of the sponsor and inspectors designated by health authorities who will be allowed access to the database, original documents, and other study files. Audit reports by the sponsor will be kept confidential.

The current legislation will comply with the terms of protecting the confidentiality of data (Regulation UE 2016/679). For this, each patient will receive an alphanumeric identification code in the study that will not include any data that allows their personal identification. The principal investigator will have an independent list that will allow the connection of the identification codes of the patients participating in the study with the clinical and personal data. This list will never leave the center. Only the researchers responsible for the study and listed in the clinical protocol approved by the Ethics Committee, as well as the competent authorities, will have access to the study data.

The trial sponsor and the researchers of the study declare his or her willingness to make the results of the trial public, preferably through scientific means of dissemination, that is, publications submitted to the peer review process.

### 2.16. Limitations of the Study

One of the variables that will be measured is usability. It must be borne in mind that part of the patients who survived a stroke do not only have disabilities but, in some cases, there are older patients who are not used to the use of new technology and this could be a problem to adapting to the STARR system. For this, applications for tablets and smartphones are very easy to use, intuitive, and guide the patient without great technological knowledge. In addition, all the wearables used in the study will be Bluetooth, facilitating the transmission of data in this way.

Another limitation could be the use for end users within the health sector since the STARR system would allow patients to be monitored, as well as to perform clinical controls and changes in their medical guidelines. But it could lead to failure if the system overloads professionals’ workload and is not easy to be managed. This fact will be taken into account when implanting alarms and communication systems with the health system so that it is manageable by the professionals and really helps to control and improve the patient.

## 3. Discussion

Quality of life can be improved by self-management interventions that accomplish more than a single domain of change. The success of these interventions depends on participation levels, impairment of participant, health services’ use, health behavior, costs, participant’s satisfaction, and associated adverse events during the intervention period. However, these interventions are often difficult, time-consuming, and human resources intensive [24].

In order to ease this intervention, new technologies could help. Furthermore, there is a rapid growth worldwide in mobile phone use, internet connectivity, and digital health technology.

A Cochrane review about mobile phone messaging for facilitating self-management in long-term conditions, such as diabetes, hypertension, and asthma, was published in 2012. They found moderate evidence in the improvement of health outcome and self-management for the three conditions, but only low evidence in health service utilization in asthmatic patients [15].

In our study, we took into consideration all the aspects of stroke and its self-management. In 2016, Spassova et al. have published an RCT, a computerized phone-based lifestyle coaching prevention system, with a 6-month follow-up in 94 patients at high risk of stroke. They reported a statistical improvement in blood-pressure measures, LDL, triglyceride values, and nutritional quality (fruit, vegetable, fewer sweets). They combined remote surveillance with tailored advice by phone call, and they concluded that it is usable and effective without being able to register severe events, such as stroke or deaths [19].

In July 2017, Lui S et al. concluded in their review of the literature that mobile health is a viable strategy to enhance stroke risk factor control, improving the glycemic control and smoking abstinence at 6 months [21].

Not only the chronic patients could benefit from this intervention but also the acute and subacute ones, as Vanacker et al. showed in their article of March 2017 where a well-trained nurse controlled remotely the stroke knowledge, secondary prevention, and rehabilitation treatment [20].

According to our study protocol, we will study chronic and disabled patients who live at home. We will try to improve all the aspects of self-management due to a holistic intervention, so if effective, the results of this study will enable stroke patients and their caregivers to deal better with the everyday life obstacles of stroke, however, there is no study about the caregivers and health professionals’ point of view of the new technology-based medicine in the reviewed literature.

The intervention is expected to promote a lifestyle change, not only promoting the therapeutic physical activity, but also alcohol and smoking consumption diminution, healthy diet implementation, and improving health literacy and information management among survivors, caregivers, and health professionals.

Adherence to the treatment, both pharmacological and non-pharmacological, is expected to increase. Control of modifiable cardiovascular risk factors is envisaged to get better. In consequence, it might reduce the recurrence of secondary strokes, reduce the number of complications, reduce the number of consultations, and readmissions, all in all, it might reduce the overload and hence reduce the health systems costs.

Caregivers are also expected to obtain assistance and related care information from health system professionals during the study. It is well known that assessing the needs of family caregivers is important for health care workers in understanding problems from the caregivers’ perspectives [24]. By improving the self-management of stroke survivors, it is intended not only to empower caregivers but also to improve their quality of life.

A follow-up of 6 months may be a short time to establish statistically significant results about the stroke recurrence, detected medical problems, complications (falls, fractures, depression, pain, etc.), and deaths. However, the intention of the research team is to perform the first analysis with the results of the 6 months of intervention and, if these are positive, then to conduct a trial with 3–5 years of follow-up.

One of the limitations of the study could be a lack of adherence to the designed application since it is a population, in some cases with advanced age and with physical limitations, resulting from their disease. The usability analysis will be determinant to develop a long-term study. In any case, studies carried out in similar populations make us think that usability will not be a problem for the continuity of the study and to demonstrate the improvement of self-management and quality of life of stroke survivors and caregivers. The potential effectiveness of the STARR system lies in the active participation of patients and caregivers.

Taking into consideration that the number of stroke survivors is increasing all around the world, a large number of stroke survivors, caregivers, and also health systems could profit from this intervention.

## Figures and Tables

**Figure 1 mps-02-00050-f001:**
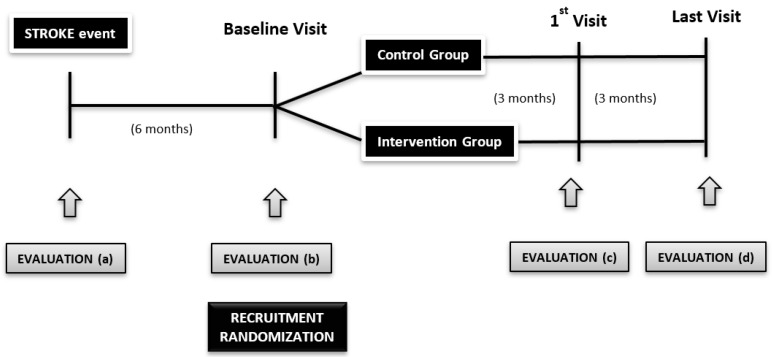
Diagram of the study.

**Table 1 mps-02-00050-t001:** Schedule of the trial variables for patients.

Stroke Survivors	Variables/Scales	Event Visit (a) Month 0	Baseline Visit (b) Month 6	First Visit (c) Month 9	Final Visit (d) Month 12
Inclusion	Age	x	x		
	Stroke characteristics and type	x	x		
	Modified Barthel Index	x	x		x
	Modified Rankin Scale	x	x		x
	Mississippi Aphasia Screening Test [MAST]	x	x		x
	Montreal Cognitive Assessment [MoCA]	x	x		x
	Gijon’s social-familial evaluation scale	x	x		
	Comorbidities	x	x		
	Charlson Comorbidity Index	x	x		
Socio Demographic	Gender, ethnic group, deprivation index, hand dominance, education level, type of job, hobbies	x	x		
Clinical/ Neurological	Cardiovascular risk factors	x	x	x	x
	Neurological physical examination: Medical Research Council Scale [MRC]	x	x	x	x
	Mississippi Aphasia Screening Test [MAST]	x	x		x
	Montreal Cognitive Assessment [MoCA]	x	x		x
	Functional Ambulation Categories [FAC]	x	x	x	x
	10 m walking test/6 min walking test	x	x	x	x
	Berg Balance Scale [BBS]	x	x	x	x
	Frenchay Arm Test [FAT]	x	x	x	x
	Asworth Modified Scale for Spasticity	x	x	x	x
	Fatigue Severity Scale [FSS],	x	x	x	x
	Line Bisection Test	x	x	x	x
	Disphagia Sensitivity Campimetry	x	x	x	x
	Pain Analogic visual scale for pain Analgesic treatment consumption	x	x	x	x
	Depression and Anxiety: Golberg Scale	x	x	x	x
	Stress	x	x	x	x
	Weight, Height, BMI, waist size, waist-to-hip ratio	x	x	x	x
	Blood pressure, heart rate, glycemia	x	x	x	x
	Need of upper limb orthoses, lower limb orthoses, and canes and wheelchair use in outdoor activities	x	x	x	x
Health Literacy	Test of Functional Health Literacy in Adults Stroke Patient Education Retention	x	x		x
Usability	System Usability Scale			x	x
Life Style	Mediterranean Diet Assessment Tool Physical activity/Exercise Toxic consumption	x	x	x	x
Blood Test	Lipidic profile (total cholesterol, high-density lipoprotein (HDL), low-density lipoprotein (LDL), cholesterol), glycemia, proteins, albumin, HbA1c, Apoprotein B, and Apoprotein A1	x	x		x
Activities of Daily Living	Modified Barthel Index (BI) Lawton Index	x x	x x		x x
Quality of life	SF-36 Stroke Impact Scale (SIS)	x x	x x		x x
Self-Management	The Southampton Stroke Self-Management Questionnaire (SSSMQ)		x	x	x
	Satisfaction Questionnaire		x		x
	Quebec User Evaluation of Satisfaction with Assistive Technology			x	x
Adherence	Post-stroke checklist		x		x
	Non-pharmacological				
	Pharmacological			x	x
Complications	Stroke recurrences Number of readmissions Number of consultations to the emergency department Number of visits/telephone calls to a general doctor Number of visits to specialist Number of secondary complications due to stroke			x	x

**Table 2 mps-02-00050-t002:** Schedule of the trial variables for caregivers.

Caregivers	Variables/Scales	Event Visit	Baseline Visit	First Visit	Final Visit
Health Literacy	Test of Functional Health Literacy in Adults Stroke Patient Education Retention	x	x		x
Quality of Life	SF-36	x	x		x
Burn-Out	Caregiver Strain Index		x		x
Self-Management	Family needs of stroke patient questionnaire		x	x	x
Satisfaction Questionnaire	Satisfaction Questionnaire		x		x
Usability	System Usability Scale			x	x

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
