# Peer review of "Efficacy and Usability of eHealth Technologies in Stroke Survivors for Prevention of a New Stroke and Improvement of Self-Management: Phase III Randomized Control Trial"

_mps, 2019, doi:10.3390/mps2020050_

Round 1
Reviewer 1 Report
The protocol is an interesting and worthwhile topic. The processes and statistics in the protocol appear to be appropriate to the study design. However the English is hard to read and substantial editing is necessary to be readable. I have attached a scanned copy of the manuscript with my written comments.

Author Response
Dear Reviewer 1,
Thank you very much for all the suggestions made.
We have corrected all the considerations made in the scanned document and the manuscript has been reviewed by a native speaker.
Sincerely
Reviewer 2 Report
This is a single center, randomized, non-blinded controlled study of the STARR system which includes wearables, decision support, predictive modeling and apps to improve self-management among stroke survivors 6 months post-stroke. The study is highly innovative.
In the inclusion/exclusion criteria, it is slightly confusing to have things as inclusion and exclusion. For example, socio-familiar dystocia and modified rankin scale are included as both. I think it is easier to understand as an inclusion or an exclusion criterion.
More information about the components of the STARR system is needed? For example, what devices will be installed in subjects homes? What are the alarms? How does it connect with the healthcare system? More information on the accuracy of wearables in measuring blood pressure would also be helpful.
Additional information about how the STARR system impacts the caregiver would be helpful
There are many primary outcomes. Please denote which primary outcome was used to power the study.
More information on how outcomes will be obtained would be helpful. The study is non-blinded but will outcomes be assessed over the phone, mail, in-person?
Minor
Would be helpful to define the timing of the acute, subacute and chronic phases
Author Response
Dear Reviewer 2,
Thanks in advance for the reviews that have improved some of the aspects of the manuscript that were not clear or needed some amendments. We think that our changes and the comments to the reviewers cover all the aspects that were raised by both reviewers. Should further comments, amendments or other explanations required, we are fully opened to clarify and/or amend them.
In the inclusion/exclusion criteria, it is slightly confusing to have things as inclusion and exclusion. For example, socio-familiar dystocia and modified rankin scale are included as both. I think it is easier to understand as an inclusion or an exclusion criterion.
We have changed the exclusion criterion following your suggestion
More information about the components of the STARR system is needed? For example, what devices will be installed in subjects homes? What are the alarms? How does it connect with the healthcare system? More information on the accuracy of wearables in measuring blood pressure would also be helpful.
We have added all the components of STARR, the alarms system, and wearables information.
Additional information about how the STARR system impacts the caregiver would be helpful
We have added a paragraph explaining the impact of the system in the caregivers, in terms of improving empowerment and quality of life.
There are many primary outcomes. Please denote which primary outcome was used to power the study.
We have detailed the primary outcome and who to obtain the sample size. We have added the bibliography of the sample size data.
More information on how outcomes will be obtained would be helpful. The study is non-blinded but will outcomes be assessed over the phone, mail, in-person?
It has been detailed in the methods section.
Minor
Would be helpful to define the timing of the acute, subacute and chronic phases
It has been defined.
Looking forward for your answer.
Sincerely,
Eunate Arana-Arri PhD
Round 2
Reviewer 1 Report
There were substantial improvements to the English conducted by the authors. However, there are additional edits that need to be made to improve the English in the manuscript. Furthermore, the improvements in the English uncovered one additional issue with the manuscript. In the manuscript it is stated that the primary outcome is the self-management questionnaire. First, the construct assessed by the questionnaire is not named (self-management can be assessed by various constructs/concepts). Second, the power analysis does not use self-management as an outcome to determine the sample size. Third, the discussion section (lines 417-420) names one of the limitations of the trial is not being able to detect stroke recurrence, implicating that this the primary outcome. This needs to be clarified in the manuscript. I have scanned in my hard copy with written suggestions for improving the English and pointing out the issue I have just outlined.

Author Response
Dear Reviewer,
First of all, thank you very much for all the suggestions that are very useful for the improvement of the manuscript.
We have added the questionaries that will be used to asses self-management of stroke survivors. This will be measured by: Southampton Stroke Self-Management Questionnarie (SSSMQ) and Family needs of stroke patient questionnaire. These two questionnaires have been validated and used in different published studies to establish the dependence of patients with stroke and to estimate self-management in this way. It has been clearly established in all sections of the manuscript that the main variable is self-management and not recurrence.
We have corrected all the additional edits that were indicated to improve the English.
Sincerely,
Eunate Arana-Arri, PhD
Reviewer 2 Report
none
Author Response
Thank you very much.
Sincerly,
Eunate Arana-Arri, PhD